# Health System Barriers to Child Mandatory and Optional Vaccination among Ukrainian Migrants in Poland in the Context of MMR and HPV Vaccines—A Qualitative Study

**DOI:** 10.3390/ijerph20010712

**Published:** 2022-12-30

**Authors:** Maria Ganczak, Paweł Kalinowski, Oskar Pasek, Łukasz Duda-Duma, Ewa Sobieraj, Jakub Goławski, Daniel Biesiada, Danielle Jansen, Johanna P. M. Vervoort, Michael Edelstein, Marta Kowalska

**Affiliations:** 1Department of Infectious Diseases, Collegium Medicum, University of Zielona Gora, 65-417 Zielona Gora, Poland; 2Department of Hygiene and Epidemiology, Medical University of Lublin, 20-059 Lublin, Poland; 3Student Research Group, Collegium Medicum, University of Zielona Gora, 65-417 Zielona Gora, Poland; 4Primary Care Clinic “Lancet”, 73-240 Bierzwnik, Poland; 5Department of General Practice and Elderly Care Medicine, University Medical Center Groningen, University of Groningen, 9712 CP Groningen, The Netherlands; 6Department of Health Sciences, University Medical Center Groningen, University of Groningen, 9712 CP Groningen, The Netherlands; 7Ziv Medical Center, Safed 7404703, Israel; 8Azrieli Faculty of Medicine, Bar-Ilan University, Safed 5290002, Israel

**Keywords:** vaccination, MMR, HPV, children, healthcare system barriers, migrants, Ukraine

## Abstract

Background Migrants’ access to healthcare services is limited. This study aimed to identify health system barriers to vaccination, specifically HPV/MMR vaccination among children in Ukrainian economic migrants (UMs). Methods Between December 2021–March 2022, a qualitative study of UMs living in Poland was conducted. Six focus groups were held with 53 UMs aged 15–45; in-depth interviews with 12 healthcare workers (HCWs) were also performed. A thematic analysis was conducted based on the six WHO health system building blocks. Results HCWs described gaps in integrating migrant status in accessible healthcare data which impeded active management of vaccination procedures. UMs reported that the digitization of healthcare services, intensified during the COVID-19 pandemic, reduced their access to primary care. Inadequate health information systems caused problems with the provision of credible vaccine information in translated forms, and language difficulties, experienced by both UMs and HCWs; this was enhanced by a lack of professional interpreting services. Although most UMs reported vaccinating children according to the Polish schedule, the variations in schedules across countries caused concern among UMs and increased HCWs’ uncertainty about how to interpret vaccination cards, particularly in the context of possible false certificates. UMs were affected by discrimination through HCWs. HPV was deprioritized by UMs due to misconceptions about non-mandatory vaccinations; the cost was also a barrier. Conclusions The study findings have implications for migrant vaccination delivery targeting children in Poland, and other UMs receiving countries. A concerted effort is required to improve UM’s awareness of the significance of vaccinations. Barriers to healthcare access must be recognized by policymakers. Importantly, removing the cost barrier may increase the uptake of the HPV vaccine among Ukrainian migrant adolescents.

## 1. Introduction

The literature suggests that some disadvantaged subgroups, including migrants and refugees, generally experience higher vaccine-preventable diseases (VPD) burden and report lower immunization coverage compared to the general population [1,2,3]. Numerous factors contribute to these findings, including poverty and difficult living conditions, which make physical distancing challenging, as well as a lack of knowledge about health and distorted perceptions about the risks posed by infectious diseases [4]. Furthermore, the above-mentioned populations are constantly facing barriers to accessing healthcare because the health systems of the hosting countries are often insufficiently responsive to their specific needs and may exclude them from vaccination plans and systems. Some important health system barriers contain language difficulties, a lack of legal entitlements to health care, a lack of outreach and community engagement capacity, a lack of cultural sensitivity, and barriers to primary care and vaccination services access, including vaccination costs. Lack of confidence in the health system of the receiving country and misconceptions about the vaccines also play an important role in suboptimal vaccination uptake among migrants [4,5,6]. The World Health Organization (WHO) has recognized this important issue and placed equity as one of the strategic priorities for immunization programs in its Immunization Agenda 2030, the global immunization strategy for the current decade [7].

Before the Russian aggression on Ukraine, Ukrainian migrants (UMs) in Poland were typical economic migrants. Mostly, they were working-aged adults attracted by employment opportunities. In the early stages, as they decide whether to stay, many fall into a pattern of migrants moving across the border and working for several months in Poland and then returning to Ukraine to extend visas before again returning to Poland. Some were making a conscious decision to leave their home country and move to Poland with the intention of settling there and becoming immigrants [6].

Ukrainian migrants (UMs) are a mobile community with low immunization rates, moving from a country with negative attitudes towards immunization and a high incidence of VPD, including measles [8,9,10,11,12,13]. Overcrowding, poor shelter and vulnerability may further put them at increased risk for infectious diseases. These large clusters of migrants coping together with a high proportion of unvaccinated people can lead to numerous outbreaks [6,10].

As long as the political and economic situation in Ukraine remains unstable and the continuous massive migration takes place, the overriding priority should be to ensure adequate medical care for migrants and refugees, including vaccinations [14]. This consequently may cause numerous public health challenges due to the need to adapt the healthcare systems to their specific requirements [10,14]. In this context, potential barriers to immunization need to be addressed.

Therefore, the primary objective of this report is to describe healthcare system barriers to MMR and HPV vaccination among Ukrainian children and teenagers in Poland. Specifically, a high influx of children refugees from Ukraine calls for urgent actions to ensure high vaccination coverage among this vulnerable population. This consequently may cause numerous public health challenges due to the need to adapt the healthcare systems to their specific requirements [10,14]. The focus of the paper is on childhood vaccination, thus convincing parents to allow their children to be vaccinated, in the context of potential barriers to immunization that their parents meet and how such barriers need to be addressed. Research shows this is different from persuading adults to accept vaccines (such as the COVID-19 vaccine) [15].

Because of the focus on healthcare system barriers, we used the six building blocks of the World Health Organization (WHO) Health Systems Framework to describe barriers: (1) service delivery: (2) health workforce: (3) information: (4) medical products: (5) vaccines and technologies, (6) financing, and leadership/governance [16]. To answer research questions, a qualitative study was conducted [17]. According to Zapata-Barrero et al., qualitative research is especially important for migration studies regarding its ability to produce comprehensive and elaborative analyses adjusted for understanding complex and interlinked challenges experienced by various migrant groups [18].

Identification of the above-mentioned healthcare system barriers will in turn help to draw conclusions from this underserved group and make key recommendations that can benefit many other countries experiencing a significant refugee influx from Ukraine, as well as other migrant groups.

## 2. Materials and Methods

### 2.1. Study Population and Sampling

#### 2.1.1. Ukrainian Migrants

A purposeful sampling technique was used aiming at maximum variation [Patton]. Due to a lack of particular physical arenas enabling the recruitment of Ukrainian migrants, the research team members gained access to a variety of migrant informants by the use of three key informants representing different characteristics (e.g., work position and education level). These three informants were asked to recruit other Ukrainian migrants with different characteristics to gain a wide range of perspectives [17]. Recruitment took place in the capital cities of two geographically different regions: in Zielona Gora (Lubuskie province), located in the western part of Poland and neighboring Germany and in Lublin (Lubelskie province), in the eastern part of Poland and neighboring Ukraine. We aimed to interview approximately 15–20 adult UMs per site (30–40 participants total) and 12–15 adolescent Ukrainian female migrants. These numbers of migrants were thought relatively easy to recruit, however, sufficient for providing adequate insight into the subject matter. Inclusion criteria were as follows: being parents/grandparents of children 0–18 years old/or being an adolescent (age 15–18 years) girl and resident in Poland for a minimum of 6 months and a maximum of 10 years (a recent migrant). Grandmothers were included in focus groups after discussions with the Board of Experts in the field from the RIVER-EU project, including a Ukrainian academic professor (a pediatrician) currently working at one of the Polish universities. A thorough medical literature review was also conducted, which found that, in some regions, grandmothers may serve as gatekeepers for health-seeking behavior, especially with regard to their daughters and daughters-in-law [18].

In each of the above-mentioned regions, an adult group of UMs included individuals representing different ages, educational levels, residence in Ukraine, command of the Polish language and length of stay in Poland. Furthermore, we purposely recruited migrants living in two geographically different regions of Poland—one neighboring Ukraine and having easy access to there regarding healthcare, another—located in the western part of Poland, about 800 km away from the Polish-Ukrainian border. Researchers aimed to assess any possible differences in migrant access to Polish and Ukrainian healthcare depending on the residence. 

Adverts for the study were circulated to institutions employing UMs, job agencies recruiting UMs, and one agency recruiting adolescents willing to study in Polish high schools; adverts were also circulated on Facebook. 

Before the focus group meeting, adult participants were given an information sheet in Polish and Ukrainian detailing the study objectives and explaining all aspects of participation; the right to withdraw from the research at any time was also clearly explained, and confidentiality was emphasized. Participants were also informed about the voluntary nature of participation and then asked to complete a consent form (in Ukrainian and Polish), including the future reuse of the data.

Regarding Ukrainian teenagers living in Poland and eligible for the HPV vaccine who were willing to be involved in the study, information on the research was sent to their legal representatives together with an invitation for an adolescent to participate. Informed consent was obtained from the adolescents and their legal representatives before the focus group meeting in accordance with the Polish Act of the Medical Profession (UZL). In Poland, participants 12–16 years of age are able to sign a co-consent. 

A short pro-forma about participants’ socio-demographic profile (age, gender, education level, residency in Ukraine, number of children—if eligible), length of stay in Poland, and number of back-and-forth travel to Ukraine per year was included. Each participant was compensated to the equivalent of PLN 100,000.

#### 2.1.2. Healthcare Professionals

In-depth semi-structured interviews were conducted with Polish HCWs involved in the delivery of vaccinations in areas with UMs. This included HCWs working in general practice, including immunization centers, and representing different job categories (GPs, gynecologists, practice nurses) and duration in practice. Recruitment focused on Zielona Gora and Lublin. We aimed to interview approximately 6 HCWs per site (12 HCWs total). This number was considered achievable and acceptable to gain insight into the subject matter; similar numbers of HCWs were recruited in other qualitative studies exploring the process of vaccinating migrant populations [19].

HCWs were identified via general practices. Potential participants were given an information sheet detailing the study objectives. Participants were informed about the voluntary nature of participation. Informed consent was signed by every HCW before an interview.

### 2.2. Data Collection

Three topic guides were developed: for adult UMs (focus groups), for adolescent girls (focus groups), and for HCWs (individual interviews) following a literature review and consultations with project team members who used a similar approach in the past [5,6,20,21]. The topic guides for UMs included semi-structured questions, such as knowledge, attitudes, perceptions related to childhood vaccination, and MMR and HPV in particular, experience with vaccination, and health system barriers to vaccination. The topic guides utilized to conduct interviews with HCWs included questions about perceived vaccination attitudes, and uptake and service delivery within the migrant population. The topic guides were checked with members of the Ukrainian community, as well as with HCWs serving this population.

During focus group discussions UMs were interviewed in person in the meeting rooms at the universities’ venues. UMs were interviewed in Polish. However, interpreters were present during all 8 meetings. Focus group discussions lasted 60–90 min. UMs were asked about their vaccination and related primary health care experiences and also requested service improvement suggestions. 

HCWs were interviewed in person in workplaces or at the universities’ venues; interviews lasted 45–60 min. HCWs were interviewed about vaccination service delivery to Ukrainian patients and their children and requested service improvement suggestions. 

Interviews were audio recorded, pseudonymized and transcribed verbatim by the team members and then checked by the professional translator for clarification and amendment.

### 2.3. Data Analysis 

Interviews were collected by the research team and then analyzed thematically using the six steps defined by Braun and Clarke (2006) [22]. Interviews were coded using preliminary work for the RIVER-EU Work Package 3 Delphi. The aim of the Delphi was to reach consensus, per country, on the significance of health system barriers in causing low vaccine uptake in the target populations and the feasibility of addressing these. This was done using a ranking method for both questions and with three rounds of questions. We followed the Guidance on Reporting and Conducting Delphi Study (CREDES) guidelines for the important steps and procedures to be taken in our study [23]. 

Thirteen barriers have been identified:
Barrier 1: Access to vaccination services is limited.Barrier 2: Language difficulties.Barrier 3: Insufficient coordination of health services.Barrier 4: Healthcare professionals are insufficiently trained and skilled in providing tailored care and information.Barrier 5: Vaccine hesitancy among healthcare professionals.Barrier 6: Lack of healthcare resources.Barrier 7: Lack of or insufficient delivery of information.Barrier 8: Lack of awareness-raising initiatives about vaccine-preventable diseases.Barrier 9: Lack of government intervention to address the influence of anti-vaccination movements.Barrier 10: Vaccinations and/or additional costs are too expensive.Barrier 11: Insufficient vaccination regulations.Barrier 12: Insufficient leadership or governmental coordination in guaranteeing the quality and availability of vaccination programs and promotion of these programs.Barrier 13: Insufficient governmental regulations in registering migrants for health care services, including vaccinations.


For the purpose of the Polish part of the project, to enhance the rigor of the analysis, coding approaches and data interpretations were discussed between Polish team members: MG, PK and MK.

The use of these 13 barriers helped to identify where to focus policy and practice recommendations. 

The themes identified were then mapped to the six WHO health systems building blocks, and the results were described according to these building blocks. Each building block contains information from the different kind of interviewees (HCWs and parents/grandparents and/or teenagers), as well as combines the vaccines enquired about, specifying whether the barriers identified are vaccine specific (i.e., only applicable to MMR and/or HPV) and more generalizable. Each of the 13 barriers described above was mapped to one or several health system building blocks [15]. It was therefore possible for the same barrier/enabler to appear multiple times across several building blocks. 

To enhance the rigor of the analysis, coding approaches and data interpretations were discussed between team members unless consensus was reached with regards to themes and how they mapped to the WHO building blocks. During the whole study, data analysis was a continuous and cyclical process of going back and forward through the data. Deductive and inductive approaches were used to analyze which health system barriers could be identified and which themes emerged from the data.

Use of the above-mentioned barriers helped to identify where to focus policy and practice recommendations. The results were described according to these building blocks.

## 3. Results

### 3.1. Participant Demographics

#### 3.1.1. Ukrainian Migrants 

Six focus group discussions (FGD) were carried out with adult UMs (4 December 2021 to 28 January 2022), three—in December 2021 in Zielona Gora (with 7; 3 and 6 participants respectively) and another three in January 2022 in Lublin (with 6 participants each), as well as two FGDs with adolescent Ukrainian girls on 23 January, 2022 in Lublin (with 6 participants each). The mean age across the study sample was 32.1 years; 44 of 45 participants (97.8%) were females. The mean duration of stay in Poland was 3.6 years, with 39.1% participants who had resided for ≤2 years in Poland, 41.3% for 3–5 years, and 19.6% for >5 years. Multiple levels of education were represented among participants; all 12 teenage girls were high school students, 30.4% UMs were high school graduates, 43.5% were bachelor’s or master’s degree holders. More than half of participants (52.2%) were coming from Ukrainian cities > 150,000 inhabitants, more than one third (34.8%) from cities ≤ 150,000 inhabitants, the rest were villagers. Regarding 34 adult UMs, all were having children; 8—1 child, 18—2 children, 4—3 children, 4—4 children; 4 of the participants were grandmothers. Around one third of participants were living in the Polish city of Zielona Góra, the rest—in Lublin. Participant demographics are further described in Table 1. 

#### 3.1.2. Health Care Workers

We carried out 12 interviews with HCWs (17 February 2021 to 11 May 2022), 6—in February 2022 in Zielona Gora and 6 in April–May 2022 in Lublin. The mean age across the study sample was 39.6 years; 8 (66.7%) participants were female. There were 10 physicians: 9 GPs and one gynecologist, and 2 nurses from the immunization centers. 

### 3.2. Health System Barriers Pertaining to the Uptake of Childhood Immunizations

Figure 1 presents identified barriers pertaining to the uptake of MMR and HPV childhood immunizations in correspondence with the relevant WHO health system building blocks.

#### 3.2.1. Building Block 1: Service Delivery 

Insufficient or limited access to vaccination services

UMs were critical regarding registering with Polish GP practices., Dissatisfaction was commonly noted around the fact that a specific identification number was required at the point of vaccination centers, which UMs not always have. 


*“We did not have a PESEL number [an analogue of the Ukrainian identification number] when we came to Poland and although I had health insurance and worked legally, without this ID they did not want to register us with a GP surgery”.*
(mother, age 35, 2 children)

Participants remained unaware that mandatory child vaccines can be freely accessed in Poland, without any additional costs covered by parents. They mentioned fears over being charged if they present for a vaccine.

*First of all, I was shocked that I didn’t have to buy anything. I kept asking: “maybe this, maybe that” and a GP looked at me strangely and said “No, we have everything here, everything is free of charge”*. (mother, age 36, 2 children)

Without a prior understanding of vaccination delivery in Poland, UMs based their expectations on intrapersonal knowledge and experiences in Ukraine, where patients make unofficial payments for health services they receive, including vaccinations. One of the participants, who had been residing in Poland for 6 years, raised the issue of bribery. 


*“In Ukraine, I do not trust doctors because all take bribes. It is very difficult to find a “normal” doctor”.*
 (grandmother, age 48, 2 grandchildren)

HCWs confirmed UMs’ reports of being surprised that patients in Poland do not have to personally pay (informally) in cash to physicians, or other medical personnel for health services they receive, including vaccinations.


*“Many times those who come fresh from Ukraine are surprised that when they come to vaccinate their children, they do not have to bring syringes, needles with them, they do not have to pay us some extra money for this vaccination. They are pleasantly surprised that it is on the side of the NHF, that it is all refunded”.*
(female GP, age 30)

Lack of or insufficient delivery of information

In general, most UMs reported that they were not offered, or directed towards, vaccination and broader health information in Ukrainian language. Many UMs described that they would need more information about what kind of documentation would be asked for at their GP surgeries, and felt that they currently lacked information in this area. Participants advocated more vaccination information being available in the Ukrainian language through different means.


*“Such leaflets which include information on vaccines are available in our GP practice waiting room. When I’m there I usually have a look at them, but they have never been written in Ukrainian”.*
(female student, age 17)

HCWs acknowledged the barrier that vaccination information was not available in both languages. There was also recognition that brochures mainly refer to recommended, optional vaccinations; brochures tackling mandatory vaccines are not available.


*“We have a lot of brochures in our waiting room. Unfortunately, these brochures mainly refer to recommended, optional vaccinations and are in Polish language”.*
(female GP, age 49)

All UMs reported difficulty finding vaccination information they trusted and resorted to using a variety of unregulated sources such as Google searches and social media, specifically Facebook and WhatsApp groups. Interestingly, many UMs were critical regarding the quality of information available in the internet.


*“You can read a blog, you can read an article, but professional articles about vaccinations in Ukrainian… I have not seen much in the Internet”.*
(mother, age 45, 2 children)

In general, those feeling most uninformed due to a lack of understandable, clear, official sources of information seem to rely more on word-of-mouth or social media for subsequent information on the ongoing vaccination programs.

Many Ukrainian parents/grandparents were currently hesitant about accepting the HPV vaccine for their adolescent daughters or granddaughters. In fact, this was the dominant vaccine that they reported struggling with in the vaccination decision-making process, which involved the evaluation of perceived potential benefits and risks. They stated that Ukrainian schools have not provided any education on this subject and that the concerns regarding vaccine safety is not addressed. 


*“Immediately after vaccination the child may feel good, and then she may be 20 years old and have a problem to get pregnant”.*
(mother, age 40, 2 children)

*“My daughter was small then, but they rather said not to vaccinate her [against HPV] in Ukraine, because later such vaccinated children might not have children. There was a lot of it [information] and it was rather negative, something like “This is dangerous!*” (mother, age 36, 2 children)

Therefore, interviewed parents and grandmothers, potential decision makers, were not knowledgeable enough to make an informative decision regarding vaccination. They stated they would need more information before making their decision, preferably in Ukrainian, on potential side effects of the vaccine, vaccine contents, summaries of clinical trials data, and when and how teenagers would be invited for a vaccine. 


*“I would like to learn more about HPV, get some details, listen to the neighbors and then decide whether to vaccinate my granddaughter”.*
(grandmother, age 48, 2 grandchildren)

Most of them indicated to be most satisfied with the information provided by their doctor.


*“The only way to convince me to get vaccinated is to tell me I’ll benefit personally. If my GP says, you know, something like: “It makes it less likely that you will get sick. Less likely that you will die”.*
 (mother, age 38, 2 children)


*“Ukrainian people are difficult to convince them to get this [HPV] vaccine. We do not trust lectures and videos on the Internet too much, nor leaflets and brochures. Our GP would be the most reliable source of information”.*
(mother, age 32, 1 child)

Many UMs felt they had not had access to sufficient understandable information about the HPV vaccine, with language barrier often brought up as an issue. One female stated she felt confused about where and how to vaccinate her daughter against HPV.

*“It’s just with this HPV vaccine that I’ve got a problem. GP surgery is directing us to school; the school is directing us to some other institutions. As a result, I don’t know who the person I need to report to is…*” (mother, age 43, 2 children)

Language barrier

All UMs mentioned language as one of the main barriers to their access to health services, specifically for newcomers. 


*“The biggest barrier is language and if it wasn’t there, everything would improve”.*
(female student, age 16)

UMs having a poor command of Polish used various coping strategies to obtain access to health-related information, such as asking fellow migrants who speak Polish for help during vaccination appointments. HCWs frequently used of medical terminology and jargon; this was the factor UMs particularly struggled with. 


*“My friend usually asks me to go with her and her son to the vaccination center because she is afraid that she will misunderstand something important”.*
(mother, age 35, 4 children)

Although most of the UMs used the help of other Polish speaking migrants, some tried to cope on their own. 


*“The language barrier is still the hardest to overcome for us. Medical terminology in Polish language differs much from Ukrainian. I had four child’s immunization appointments, and every time I had to translate lots of medical terms with the help of Internet”.*
(mother, age 25, 2 children)

Communication barriers during consultations were also reported by HCWs. To overcome language barriers, most interviewed HCWs reported using online translation tools to aid communication. 


*“Our first main concern is the language barrier, we speak no Ukrainian at all. Some people come with a friend or family member who translates. They [patients] type in there and give me the phone. I read this and then… we try to communicate in some way”.*
(female GP, age 49)

Some HCWs reported relying on colleagues with Ukrainian language skills, including multilingual physicians, nurses or receptionists, to translate documents and to help during vaccination appointments. Both parties felt that language barriers still created uncertainties around messages lost in translation. 


*“It is often the case that there are such comments: “I don’t know how to say it”, “I don’t know how to describe it”, “what is it called?”. It takes time to get to what is going on, you know, and there are difficulties to communicate correctly”.*
(male GP, age 30)

There is also high risk of misinterpretation; it was clearly expressed by one of the GPs.


*“There may be a large field of error when translating, for example, some side effects, or when exactly they [UMs] should report for the next dose, how to prepare for vaccination according to the schedule, etc.” *
(male GP, age 28)

Virtual consultations due to closed surgeries were perceived by HCWs as challenging because of a feeling of not being understood or—less often—with them misunderstanding patients.


*“In the era of telemedicine, communication [with a Ukrainian patient] is difficult. If I have a patient in the office, it is somehow easier to get along, and yet somehow on the phone it is more difficult. I’m usually the one who is misunderstood. When there is a lot of information to provide, I think this is the most difficult thing for us”.*
(female GP, age 30).

#### 3.2.2. Building Block 2: Health Workforce

Lack of healthcare resources

HCWs reported that for the general Polish population within the time assigned per vaccination appointment (approximately 10–15 min) it is difficult to provide vaccine information, examine a child and document vaccine delivery. In the case of UMs, this is even more challenging because of common communication barriers. 


*“The more children are registered with a GP surgery, the less time is left to talk to their parents. I think there’s a problem here, that we can’t spend much time to explain to them the issues related to vaccination”.*
(female nurse, age 29)


*“Every doctor is obliged to inform about recommended vaccinations, but it is well known that technically it is difficult due to the lack of time”.*
(female GP, age 58)

Lack of interpreters

During the FGDs and interviews, participants (UMs and HCWs) repeatedly commented on the need of the commonly accessible, free services of an at a site interpreter.


*“Sometimes we called the GP surgery and asked if there was a doctor who understood at least Russian, so that we could get along with him.…”*
(mother, age 33, 1 child)


*“We do not have the support of interpreters. Google on my computer serves as a translator”.*
 (female GP, age 32)

A proposed option was to create a hot-line operated by a Ukrainian-Polish speaking interpreter but there was recognition that cost could be a barrier regarding both: at the site interpreters and the hot-line.


*“It would be fantastic to have a hot-line. Ideally, it should be operated by a Ukrainian doctor who works in Poland. Because then he will explain everything thoroughly to us”.*
(mother, age 45, 2 children)

Discrimination by HCW against the target population

Poor treatment by medical staff was reported during registration processes and/or vaccination appointments. UMs reported being judged unfairly or treated with disrespect. For example, two highly educated female migrants in their 30 s claimed that they felt disregarded by the doctors merely for being migrants.


*“Some doctors have something against Ukrainians. It’s just that there’s this type of attitude. “Stupid Ukrainians, they don’t understand anything...”—that’s how it was said”.*
(mother, age 26, 2 children)


*“There are cases where Ukrainians are treated worse than Poles: “you take away our money, our work...” Personally, I would like to be treated in the same way as Poles”.*
(mother, age 29, 1 child)

There were also some other examples of participants that described being discriminated. 


*“There are doctors who behave in such a way that you feel as if you’re worthless”.*
(father, age 49, 1 child)


*“The doctor misread my name and then said that “you [Ukrainians] have come here and now we have a problem with you. ”I felt kind of weird”.*
(mother, age 43, 2 children)

#### 3.2.3. Building Block 3: Health Information System

Both, UMs and HCWs reported that in some cases vaccination records were incomplete, lost or falsified. This may pose a challenge to correctly identify missing vaccines. 


*“We [medical staff] cannot enforce the vaccination record in any way if they [UMs] do not want to provide us with documentation”.*
(female nurse, age 30)

Of note, some UMs stated that in Ukraine, false vaccination certificates can be readily obtained by bribing HCWs. Therefore, some UMs might obtain fake immunization records for their children; this issue was also repeatedly raised during our interviews with medical staff. 

*“As a doctor, I can expect some children [UMs] to be unvaccinated. Even if they have evidence of vaccination, such* vaccination *record cards can easily be forged”.*(female GP, age 45)

#### 3.2.4. Building Block 4: Medical Products

Optional status of HPV vaccine

UMs made clear distinction between mandatory vaccines, such as MMR, and “optional” ones such as HPV. The latter one is not universally offered in the Ukrainian immunization program and its cost has to be covered by the patient himself [10,24]. This in turn may make UMs consider it a less important vaccine, which may negatively influence uptake. 


*“I have not heard that someone in Ukraine was vaccinated with this [HPV] vaccine”.*
(mother, age 42, 3 children)


*“Nobody asked me about the HPV vaccine and I’m afraid that Ukrainian migrant women have little knowledge about it...”*
(male gynecologist, age 60) 

Lack of trust in the quality of vaccine

Safe, fast, equitable access to vaccination must remain a priority. However, UMs and Polish HCWs reported the periodic lack of certain vaccines in Ukraine.


*“We had a problem with the Hepatitis vaccine, it was called a “dangerous” vaccine. They [medical personnel] did not store it properly, I remember a report on TV when they commented on it. Later, this vaccine was contraindicated and the son could not receive it. I waited, but then there was no vaccine for a long time”.*
(mother, age 35, 2 children) 

There were also commonly held views among UMs that vaccines administered in Ukraine are of doubtful quality and have poor effectiveness and that vaccination in Ukraine could cause illness, even death. Several UMs also reported concerns that having the vaccine could cause harm.


*“I’ve heard that one child didn’t get up on his feet after the vaccine”.*
(mother, age 42, 3 children) 


*“In my little town, some children died after the diphtheria vaccine”.*
(mother, age 32, 1 child)

#### 3.2.5. Building Block 5: Financing

Direct cost of the HPV vaccine

Confusion arose for the HPV vaccination as a non-mandatory, self-paid vaccination, which was reported as rarely administered in Ukraine. In Poland, as in some other eastern and central European countries, including Ukraine, there are mandatory vaccinations that are included in the National Immunisation Programme (NIP) and provided at no cost to all children, including migrants [25]. There are also several vaccinations, such as HPV vaccination, recommended in the NIP. However, regarding HPV, 50% of its cost has to be covered by a patient [25]. Participants reported that self-paid vaccines for children are not of interest for people living in Ukraine because such vaccines are perceived as too costly.


*“Only few people will decide on such a vaccine [HPV] in Ukraine because they are simply poor and will not spend money on vaccinations”.*
(mother, age 43, 3 children)

It was also a common argument brought by our participants that there is no uptake of self-funded vaccines, such as the HPV vaccine, among UMs in Poland due to their high cost. Although most of our participants have not heard about the HPV vaccine, or their knowledge regarding this topic was very poor, after obtaining adequate information during the FGs meetings, some would also consider it, if the cost was fully refunded.


*“If we would not have to pay for this vaccination [against HPV] then we would probably accept it”.*
 (female student, age 17) 


*“I think that in the future HPV vaccination will be free of charge in Poland. And then, Ukrainian females will start to get vaccinated”.*
(mother, age 43, 3 children) 

Only one out of 42 female migrants declared she would consider HPV vaccination for her daughter. She said she would like to take the opportunity of a 50% refund; in Ukraine, 100% of the cost of the vaccine has to be covered by a patient. 


*“I would like to vaccinate my daughter. I would do it, because in Ukraine this vaccine costs so much... We don’t have such an option, as in Poland, that the state pays half of the price”.*
 (mother, age 37, 3 children)

Similarly, all HCWs reported that in general, UMs are not interested in recommended, self-paid vaccinations. HCWs clearly stated that for UMs any additional cost of the HPV vaccine which they would have to cover (around 140–160 PLN per dose) is a problem that is difficult to overcome. 


*“When it comes to recommended vaccinations, like HPV, there is no interest. It is not common that someone from migrant communities asks about recommended vaccinations”.*
 (female GP, age 58)

#### 3.2.6. Building Block 6: Leadership and Governance

Insufficient coordination of health services (between Poland and Ukraine)

Vaccinating migrant children in their country of origin could cause disturbance of the hosting country immunization schedule due to potential variations in vaccines and scheduling between national programs. Both UMs and HCWs noted that these variations led to some uncertainties. 


*“GP understood what was written in the vaccination record. But it turned out that there were 2 doses in the Ukrainian schedule, however, according to Polish schedule we need 3 doses. And I’m kind of confused”.*
 (mother, age 40, 2 children)

In addition to the polio vaccine, variations in vaccines and scheduling between national programs include Diphtheria and Tetanus, Pertussis and Hib vaccines [24,25]. 


*“Usually single doses are missing in their [UMs] vaccination records. For example, we have to administer an extra dose regarding pertussis vaccine, because there are fewer doses recommended in Ukraine when compared to our country”.*
 (female GP, age 49)

Moreover, Ukraine is yet to introduce pneumococcal and rotavirus vaccine, introduced in Poland in 2016 and 2020 respectively [24,25,26]. Such additional vaccinations, administered within a short space of time, were also reported as a challenge by Ukrainian parents. 

Interviewed HCWs clearly stated that although there is no official regulation, they routinely ask UMs for a sworn translation of child’s vaccination record.


*“If it is about foreigners, we ask to translate vaccination records because we have problems in translation. This must be translated by a sworn translator, because it refers to medical terms. With the translated documentation, it is easier for us to check all vaccinations received by the child”.*
 (female GP, age 30) 

However, access to Ukrainian-Polish sworn translations may currently be difficult. UMs reported that in some places translation services are not efficient due to the long waiting periods; furthermore, professional translation services are costly. 


*“We were looking for a sworn translator and finally found one but there was a long waiting period for his service and we finally gave up. So we still don’t have this translation. That’s why I’ve got a problem with kids’ vaccinations”.*
(mother, age 37, 3 children) 

Furthermore, at an institutional level, HCWs reported they faced challenges in determining which vaccines had been administered to the child, with many UMs entering Poland without any proof of vaccination status. 


*“They don’t always bring vaccination records, sometimes it’s just personal information”.*
(female GP, age 45) 

Lack of awareness-raising initiatives about vaccine-preventable diseases

Most UMs pointed out the lack of informative campaigns regarding HPV vaccination in their country; this may result in a lack of awareness of the need for immunization.


*“In Ukraine, you have to look for information about this vaccine [HPV] yourself. No one will tell you that it’s needed”.*
 (mother, age 26, 2 children)


*“There are no media campaigns in Ukraine regarding this [HPV] vaccine. If you want it for yourself, you have to pay quite a lot of money to get it. That’s how it works”.*
 (mother, age 35, 2 children) 

Still, not much is offered to address the above-mentioned issues in Polish public space; this was expressed by our respondents as follows: 


*”Since we have been in Poland, I have not heard about this vaccine [against HPV]”.*
 (mother, age 43, 2 children)

Since 2012, free HPV vaccination is additionally offered by some local government prevention programs [27], however, schools are not provided with curricula designed to teach adolescents that the HPV vaccine is cancer prevention. UMs complained that there are no awareness programs for school students. 


*“Recently, as a mother of a 12-year-old girl, I received a letter from the school, where it was written that Zielona Góra—as a city—is refunding the vaccine against HPV. That was the first time in my life I learned about this vaccine”.*
(mother, age 40, 2 children) 


*“In the [Polish] school we attend, there are no classes explaining the meaning of HPV vaccination”.*
 (female student, age 17) 


*“There were no lessons about HPV vaccinations in my Ukrainian or Polish school”.*
 (female student, age 15)

In addition, interviewed HCWs admitted that HPV vaccine posters are not available in GP surgeries’ waiting rooms, exam rooms, etc. 


*“In our vaccination center there are leaflets about various vaccinations available in the waiting room, but not about HPV”.*
(female nurse, age 29)

Generally, regarding the HPV vaccine, there were no significant important differences observed between the findings from adolescent focus groups and the ones with parents.

HCWs also mentioned lack of awareness-raising initiatives about vaccine preventable diseases as one of the main barriers to increasing HPV uptake among adolescent Ukrainian migrants.


*“It is not intended that the National Health Fund finances institutionally preventive activities, such as health promotion, at the GP practice level. There is only secondary or tertiary prophylaxis—only when something wrong happens”.*
 (male GP, age 60)

Lack of government intervention to address the influence of anti-vaccination movements

No major religion opposes vaccines in Ukraine—however, members of several religious groups have in certain instances, come up with religious justifications for vaccine refusal or used their authority to turn people off immunization. UMs described the above-mentioned issue as follows:


*“Well, there are religious groups that refuse vaccinations at all. Then, there is a problem because their kids start to get sick, a problem in kindergarten, school, in contacts with other children…”*
(mother, age 40, 2 children)


*“There are these numerous groups on the internet, on television, that are against vaccination. They have different opinions because of religious beliefs”.*
(mother, age 33, 4 children)

Some Ukrainian HCWs may be hesitant about getting children vaccinated due to vaccine hesitancy expressed by Ukrainian HCWs. 


*“When I gave birth, the doctor urged me not to immunize her for hepatitis B”.*
 (mother, age 35, 2 children)

Unresponsiveness and passivity of the system

Soon after UMs and their families arrive in Poland, they should be supported to register with a GP practice and attend a “new patient consultation” to assess their care needs, including vaccinations. However, several interactions with the National Health Fund (NHF) before GP registration should take place. At a minimum, this means obtaining ID [PESEL]. The lack of ID is a barrier to patient registration.

Although UMs with a residence permit have to be affiliated to the NHF, registration with GP is an independent, sovereign decision. Therefore, the registration process with GP could be delayed for months or even years as considered less important than other life needs. Separate legal regulations how to reach this hard-to-reach population have not yet been published. Most HCWs felt this is an important barrier to child vaccination.


*“How to enforce the vaccination card? If the mother does not enroll her child to any GP surgery or the child is registered with private health care, such a child disappears from the vaccination registry system”.*
(male GP, age 30) 


*“How can a nurse from the vaccination center find out that a new person from Ukraine has suddenly appeared? There are no governmental regulations regarding registration of migrants for health care services, including vaccinations”.*
 (female nurse, age 31)


*“Whether they come to the vaccination or not, we do not have any influence on it”.*
 (female nurse, age 29)

## 4. Discussion

This study identified the main barriers experienced by Ukrainian economic migrants in accessing and utilizing Polish vaccination services with the use of the WHO health systems building block framework. 

Interestingly, as Figure 1 shows, two general types of barriers to vaccination were observed: resource challenges and non-resource challenges (e.g., incompatible vaccine schedules, discrimination, HPV’s status as optional, lack of trust. Although non-resource barriers are more challenging and time consuming, the solution to resolve resource problems is-to a certain degree-fairly straightforward. Having more health system resources available and making better use of the resources are two approaches that can suffice the needs of HCWs and meet UMs’ expectations. 

One of the main barriers in UMs access to health services in Poland was the lack of language competence; this barrier was difficult to overcome specifically for newcomers. UMs felt insecure and anxious about their interactions with medical staff who do not speak a familiar language at a time when they wanted to feel cared for. Digitization of healthcare services, intensified during COVID-19 pandemic, coupled with language barriers, reduced migrants’ access to information on healthcare services. Closed surgeries necessitated a reliance on virtual consultations, which were perceived by both UMs and HCWs as challenging because of experiences with misunderstanding or a feeling of not being understood. Language barrier is among the most commonly cited factors in a review of barriers to the use of health services among migrants [28]. Our findings are also consistent with studies conducted in other receiving countries [14,29,30,31]. For instance, the Czech study found that the use of GPs by different migrant groups was positively correlated with their knowledge of the language [29]. Another study from Greece documented how increased knowledge of existing health services was associated with migrants’ increased language competence [30]. Similarly, HCWs in 16 European countries described barriers related to language among migrant minorities [32]. 

Our study identified another important health system barrier, a lack of access to inter-preting services. This preceded using various coping strategies, both by UMs and HCWs. Such strategies may result in misinterpretation of information, breach of confidentiality, and safeguarding concerns and are not recommended [20,33]. Mobile phone translation apps, as well as relatives and friends cannot be relied on to accurately interpret health-related information. The latter ones are less likely to provide objectivity [20,34]. In addition, although there is no legal obligation for a sworn translator to translate a vaccination record drawn up abroad, this is commonly practiced by Polish HCWs. Of note, the translation—even by a sworn translator—of hand-kept vaccination records may not always be correct. Some important entries may be sometimes omitted from translations. Others are described as “unreadable”, due to the fact that some translators may not be aware of the specificity of the Ukrainian vaccination records [26]. Therefore, when deciding on further immunization, it should be required to rely both on the translation (if available) and on the original documentation. Sworn translation demand may cause some additional barriers to vaccination experienced by UMs, as translation services are usually not efficient due to the long waiting periods and professional translation services are costly. Examples from some other EU countries, e.g., Norway and the UK, show that migrants have a free access to an interpreter [21,21,31]. However, so far Polish government has not developed and implemented adequate policies to effectively help UMs to minimize the language barrier and improve service delivery. 

The results of our study clearly indicate that problem with access to adjusted information constitutes one of the main barriers to UMs in Poland accessing quality health care, including vaccination services. In general, this is one the most common barriers quoted in studies of migrants’ access to health services in EU countries [28,31]. Despite the use of the Internet by the vast majority of UMs to find relevant, unbiased information about the Polish health care system and child vaccination, this was insufficient in the case of migrants who were inexperienced and/or not proficient with the Polish language. Both UMs and HCWs expressed that the availability of Internet pages and leaflets in Ukrainian was limited. This is in line with the results of earlier studies conducted in countries receiving migrants [20,28,29,30,31,32]. 

Additionally, the short time routinely assigned per GP appointment impeded the smooth flow of vaccine information delivered by an HCW; common communication barriers in the case of UMs put even higher burden on health workforce in the light of time constrain. Of note, Poland has 2.4 physicians per 1000 inhabitants—the lowest ratio in the EU [34]. 

The fact that our respondents mainly accessed Ukrainian sources of information, combined with the powerful anti-vaccination movement [6,9,10,24], may result in UMs in Poland being exposed to negative views about vaccination. The UMs’ population comes from a country where myths and misconceptions about non-mandatory, self-paid vaccinations are often cited by concerned parents as reasons to question having their children vaccinated [6,10]. For reasons mentioned above, providing professional information about the HPV vaccine, both on the national and school level, would be of great value [35]. Sadly, currently there is a lack of HPV-raising awareness campaigns in Poland.

Furthermore, the lack of adequate information about variations in childhood vaccination schedules across two countries and the number of catch-up vaccinations that should be administered within a short space of time caused concerns both among UMs and HCWs. For the latter ones, the phenomenon of fake child vaccination records in Ukraine posed another challenge. In Ukraine, giving bribes to doctors, although illegal, was described as quite common [24,36,37]. In a nationally representative household survey, 57% of respondents reported they have ever personally paid informally in cash to physicians, medical staff or other personnel in healthcare facilities [36]; 70% of patients make unofficial payments for health services they receive, including vaccinations [38]. According to the current Polish national recommendations, it is correct asking the caregiver whether the child has definitely received all the vaccines entered in the documents. However, it is not clearly stated which procedure to follow in a case where during the visit for vaccination the child’s guardian states that he was not actually vaccinated, despite the entry in the documentation [24]. Unfortunately, the absence of the Electronic Vaccination Data System in Ukraine does not allow the medical staff to effectively obtain information on vaccination history; this hampers international coordination of vaccination services. The aforementioned issues are likely to affect Ukrainian migrant populations in other countries. The lack of adequate health/vaccine information for migrants highlights the failure of the current policies and initiatives on migrant health in Poland; both could negatively influence migrants’ access to vaccination services.

Another barrier related to the organization of health care was that UMs who come to Poland have to obtain a personal number to get free access to healthcare. Without this UMs have not the right to choose a GP. We also found that vaccination and healthcare experiences in Ukraine shaped expectations of services in Poland. UMs reported they had experienced particularly high costs of paid health services in Ukraine. Therefore, they were concerned about additional costs associated with vaccines and vaccination in Poland. This points to a significant barrier with dissemination of correct information to migrants, as payment is not required at the Polish vaccination centers. This exemplifies a general lack of knowledge among migrants about their entitlement to health care, which is the case for several other EU countries [20,28,31]. 

As described in the results, Polish policymakers have not designed a clear and coordinated system for ensuring that migrants are vaccinated. A foreigner residing in Poland is subject to the mandatory NIP after 3 months of stay. Interviewed HCWs described that migrants are being identified when contacting GP surgeries by themselves or with their children and then being referred to mandatory vaccinations if there is such a need. However, most HCWs stated that migrants who are lacking vaccinations are likely not to be identified when not accessing GP clinics. Studies show that persons who choose to migrate have better health than the native population [39], therefore may not use heath services. 

A review of barriers experienced by migrants when using health care services noted that discourteous care and stereotypical attitudes towards ethnic minority patients can act as a barrier. As shown in our study, feelings of being treated differently than the majority Polish population evoked emotional distress among some UMs. Migrants felt disregarded and discriminated. In addition to UMs’ experience of other barriers, a sense of discrimination may cause them to avoid using Polish health care services. Appropriate verbal and nonverbal communication styles are important to avoid feelings of discrimination and stigmatization [40].

The “financing” WHO block remains the crucial input component to the health system, specifically, vaccination. In Poland all mandatory vaccines listed in the NIP are free for infants, children and adolescents, including migrants; however, there may be fees for recommended vaccines [25]. It was mentioned by the vast majority of interviewed HCWs that additional vaccinations were overwhelmingly not accepted by Ukrainian parents since they don’t receive any financial support from the state and most vaccines are expensive. This also refers to the HPV vaccine. Although the state partly refunds the product, the price migrant parents need to pay for this vaccine would likely be a significant financial burden in contrast to relatively low salaries and difficult living conditions. The latter factor is also connected with constant financial support which UMs in Poland provide regarding those currently living in Ukraine. Alarmingly low uptake of self-funded vaccines among migrants in other EU countries was commonly reported by other authors evaluating this issue [19,20,31,41]. 

Several recommendations can be implemented according to this study results. System-level policy recommendations include improvement of government’ leadership and governance to recognize, assess and address the needs and priorities of UMs resulted in breaches in developing and implementing effective policies and interventions to help UMs better utilize Polish vaccine delivery system. In addition, adequate awareness-raising initiatives taken by Polish policy makers are needed to improve UMs access to correct and biased information, specifically regarding vaccine importance and safety. Full reimbursement of HPV will help to reduce the financial barrier for vaccination. Adequate health information systems would help with the provision of credible vaccine information in translated forms, and minimize language and communication difficulties, which were experienced by both UMs and HCWs. 

Our research highlights the limitation of the current national migrant registry in Poland and the need of integrating migrant status in accessible healthcare data. Such a policy could arm GP clinics personnel to actively manage child vaccination procedures. HCWs overwhelmingly agreed that without a clear protocol or guidelines on how to deliver mandatory child vaccines to migrant children, vaccination might not be offered to those in need.

The fact that our respondents mainly accessed Ukrainian sources of information, combined with the powerful anti-vaccination movement [6,9,10,24], may result in UMs in Poland being exposed to negative views about vaccination. The UMs’ population comes from the country where myths and misconceptions about non-mandatory, self-paid vaccinations are often cited by concerned parents as reasons to question having their children vaccinated [6,10]. For reasons mentioned above, providing professional information about the HPV vaccine, both on the national and school level, would be of great value [35]. 

Provider level behavior changes are also needed. Currently, in Poland there are no courses on migrant health for GP clinics’ personnel. Such courses could positively impact unfriendly HCWs’ attitudes. In a recent study, over two-thirds of those experiencing poor treatment had a disruption in care as a result, including changing providers, delaying or forging needed care, or not following the provider’s recommendations [42]. Other studies highlighted the importance of cultural competence training as a means of eliminating racial/ethnic disparities in healthcare [43]. In this context, the emphasis should be put on government role to ensure equal treatment of all patients presenting to vaccination services, together with a an unrestricted attitude towards UMs among HCWs. 

### Strengths and Limitations 

The strength of this study is its innovative approach. The use of the WHO health system building blocks framework allowed us to look at health system barriers to vaccination from different perspectives. The inclusion of the perspectives from community members and HCWs, and the triangulation of data strengthened the results. A range of migrants were recruited, along with HCWs representing different professions. In addition, to improve the reliability, interviews were analyzed by various team members; all stages were discussed regularly between them to improve quality and consistency. 

Although we assume that data saturation was reached in our samples, we are conscious that UMs living in some other parts of the country may have different experiences with the Polish health system. This core study weakness was also discussed by other authors who conducted qualitative studies [44]. However, the fact that our study was planned in two geographically different regions of Poland might reduce this bias.

Focus groups with the migrants occurred before the Russian invasion of Ukraine, however, provider interviews occurred before and also after the invasion. The invasion and influx of refugees additionally strain the healthcare system, and this may also influence providers that we spoke to.

Another weakness integral to the FG format, which makes the results obtained harder to generalize to the larger population, is its participant snowball selection system. However, UMs differed regarding education level, residence, length of stay in Poland and command of Polish language which meant relatively heterogeneous opinions could be obtained. 

## 5. Conclusions

We found that the WHO health system building blocks framework contributed well to assessing health care related barriers experienced by UMs in Poland, specifically barriers regarding MMR and HPV vaccinations. The above-mentioned blocks were influenced by numerous factors and strongly interlinked with each other. These included cross-cutting components such as leadership/governance and health information systems, key input components to the health system, specifically, financing and the health workforce, as well as the immediate outputs of the health system, namely access to essential medical products and service delivery. This novel approach could also be relevant for understanding other migrant groups. 

Several barriers to vaccination that were identified were in line with the WHO framework; they depended on the variety of factors, including the migrant status, command of the Polish language, years of stay in Poland. Interestingly, Figure 1 shows two general types of barriers to vaccination: resource challenges and non-resource challenges (e.g., incompatible vaccine schedules, discrimination, HPV’s status as optional, lack of trust. Although non-resource barriers are more challenging and time consuming, the solution to resolve resource problems is-to a certain degree-fairly straightforward. Having more health system resources available and making better use of the resources are two approaches that can suffice the needs of HCWs and meet UMs’ expectations. 

The lack of government leadership and governance to recognize, assess and address the needs and priorities of UMs resulted in breaches in developing and implementing effective policies and interventions to help UMs better utilize the Polish vaccine delivery system. Gaps in integrating migrant status in accessible health care data impeded active management of child vaccination’ procedures. Poor knowledge about VPDs and vaccines among UMs fueled by insufficient public health messaging and inadequate awareness raising initiatives taken by Polish policy makers, made UMs vulnerable to accessing confusing, incorrect or biased information; concerns were raised around vaccine importance and safety regarding recommended self-paid vaccines, including the HPV vaccine. Lack of full reimbursement of this vaccine was reported as an additional barrier to vaccination. Inadequate health information systems caused problems with the provision of credible vaccine information in translated forms, and language and communication difficulties, which were experienced by both UMs and HCWs; this was enhanced by a lack of availability of professional interpreting services. Additionally, the variations in vaccination schedules across countries caused concern among UMs and increased HCWs’ uncertainty how to interpret vaccination cards, particularly in the context of possible false certificates and the absence of the Electronic Vaccination Data System in Ukraine; these problems are likely to affect Ukrainian migrant populations in other countries and pose a challenge regarding international coordination of vaccination services. Barriers such as lack of specific guidelines and tailored, culturally sensitive knowledge of HCWs might be responsible for discriminatory access to vaccination services reported by UMs. Poor service delivery was also affected by shortage of health workforce. 

These findings have important implications for migrant vaccination policy and delivery targeting children and adolescents in Poland, and other Ukrainian migrant-receiving countries. First, barriers to healthcare access addressed in this study with the use of the WHO health systems building blocks framework must be recognized by policymakers. Second, concerted effort is required to improve Ukrainian community awareness about the urgent need for vaccination through the provision of information in Ukrainian language, media coverage, HCWs’ mobilization and community outreach programs. Finally, trust in HPV vaccine must be build up, which requires the training of HCWs to identify and address community-specific vaccine concerns, the provision of transparent and reliable information, the engagement of GPs and trusted community representatives; reimbursement of the vaccine cost should be a priority. 

## Figures and Tables

**Figure 1 ijerph-20-00712-f001:**
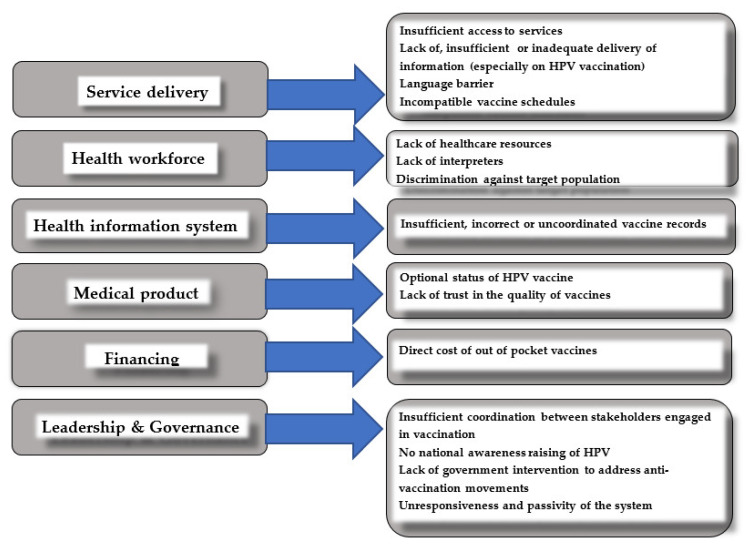
A summary of the barriers identified across UMs for each WHO health system building block.

**Table 1 ijerph-20-00712-t001:** Demographics of study participants: Ukrainian migrants (*n* = 46).

Characteristic	*n*	%
Age (years), mean: 32.1		
<18	12	26.1
18–30	7	15.2
31–40	16	34.8
>40	11	23.9
Gender		
Female	45	97.8
Literacy		
At high school	12	26.1
High school graduate	14	30.4
Bachelor/master	20	43.5
Time since arrival in Poland (years), mean 3.6		
≤2	18	39.1
3–5	19	41.3
6–10	9	19.6
Residence in Ukraine		
City > 150,000 inhabitants	24	52.2
City ≤ 150,000 inhabitants	16	34.8
Village	6	13.0
Residence in Poland		
Zielona Gora	16	34.8
Lublin	30	65.2

## Data Availability

The data underlying this article will be shared upon reasonable request, after review of the request by a RIVER-EU committee chaired by the coordinator (UMCG).

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
