# Peer review of "Health System Barriers to Child Mandatory and Optional Vaccination among Ukrainian Migrants in Poland in the Context of MMR and HPV Vaccines—A Qualitative Study"

_ijerph, 2022, doi:10.3390/ijerph20010712_

Round 1

Reviewer 1 Report

The authors showed that WHO system building blocks frameworks was suitable for assesing healthcare related barriers in the study group.

Author Response

A qualitative study on the health system barriers to child vaccination among Ukrainian migrants in Poland

REVIEWER 1:

The authors made a retrospective analysis of.

The methodology and statistical analysis, data presentation and discussion is greatly appreciated.

However some issues remain unanswered as follows:

1-Main question addressed by the research .

- language was the main barriers in UMs access to health services in Poland and the lack of access to interpreting services as another important health system barrier. This led to UMs and HCWs using a range of coping mechanisms such as bringing a friend or Google translation feelings of being treated differently than the majority Polish population evoked emotional distress among some UMs. the vast majority of interviewed HCWs that HPV, was overwhelmingly not accepted  by UMs because of the financial barrier.

 2-In the literature there is knowledge that variations in vaccination schedules across countries present.

Due to different health care policies of each country, there are disproproptionally affected vulnerable populations such as migrants. They addressed this knowledge and confirmed this. This confirmation is important for reinforcement of current knowledge in the subject area.

3-The use of the WHO health system frame work and interviews were analyzed by various team members in order to look at health system barriers to vaccination from different perspectives.

4-They stated focus group discussion format is an weak selection system and leads to heterogeneous opinions within UMs differed regarding education level, residence, length of stay in Poland and command of Polish language.

5-Conclusions are very much appreciated.

6-References are OK

7-I find all tables and figures useful.

We would like to thank the Reviewer for all valuable comments.

Reviewer 2 Report

The authors wanted to study the perceptions of migrants from Ukrain to Poland and also Health care workers on immunization services What does it add to the subject area compared with other published As Ukrain migrant is a new problem the study has more relevance in cross border health studies and human right protection issues This is a good study. The discussion on qualitative study has a style . Please see the guidelines for appraisal of qualitative research

Suggestion:

Sociogram would have been included in manuscript

Why both purposive as well as snow ball sampling not clear

Yourts with test results of  WHO building block. The study question is not that 

Author Response

REVIEWER 2:

The authors wanted to study the perceptions of migrants from Ukraine to Poland and also Health care workers on immunization services. What does it add to the subject area compared with other published As Ukraine migrant is a new problem the study has more relevance in cross border health studies and human right protection issues This is a good study. The discussion on qualitative study has a style . Please see the guidelines for appraisal of qualitative research

Suggestion:

  1. Sociogram would have been included in manuscript.

If we're right, sociograms in qualitative studies are used e.g., to acknowledge group processes and dynamics in focus groups and to explore how group facilitators influence group process and content. If our goal was to make a sociogram, we had to anticipate this during the data collection. But we didn't do that so unfortunately we can't add a sociogram.

We mentioned this limitation in the Discussion:

Another limitation is that we didn’t use a sociogram. Using a sociogram in addition to the focus group transcriptions might have provided a more balanced analysis of both the process and outcomes of the focus groups.

  1. Why both purposive as well as snow ball sampling not clear.

This was corrected as follows:

A purposeful sampling technique was used aiming at maximum variation [Patton]. Due to a lack of particular physical arenas enabling recruitment of Ukrainian migrants, the research team members gained access to a variety of migrant informants by the use of three key informants representing different characteristics (e.g work position and education level). These three informants where asked to recruit other Ukrainian migrants with different characteristics to gain a wide range of perspectives. [17].

Patton MQ. Qualitative research and evaluation methods. 3. Thousand Oaks: Sage; 2002.

  1. Yourts with test results of  WHO building block. The study question is not that. 

We would like to thank the Reviewer for this valuable comment. We have rearranged the paragraph as follows:

As long as the political and economic situation in Ukraine remains unstable, and the continuous massive migration takes place, the overriding priority should be to ensure adequate medical care for migrants and refugees, including vaccinations [14]. This consequently may cause numerous public health challenges due to the need to adapt the healthcare systems to their specific requirements [10,14]. In this context potential barriers to immunization need to be addressed.

Therefore, the primary objective of this report is to describe healthcare system barriers to MMR and HPV vaccination among Ukrainian children and teenagers in Poland. Because of the focus on healthcare system barriers, we used the six building blocks of the World Health Organization (WHO) Health Systems Framework to describe barriers: (1) service delivery: (2) health workforce: (3) information: (4) medical products: (5) vaccines and technologies, (6) financing, and leadership/governance  [15].

Identification of  the above mentioned healthcare system barriers will in turn help to draw conclusions from this underserved group and make key recommendations that can benefit many other countries experiencing a significant refugee influx from Ukraine, as well as other migrant groups.

Reviewer 3 Report

A well-designed study embedded in previous research as shown by the references. Use of the WHO framework was a good idea.

The text needs revision: missing -s in 3rd person singular verbs, missing articles, punctuation errors, spelling errors, inconsistencies in the use of italics.

See the attached annotated version of the text (but check even more thoroughly). Check also whether there are remaining double spaces.

Author Response

We would like to thank the Reviewer very much for all valuable comments and corrections.

The text has been revised: missing -s in 3rd person singular verbs, missing articles, punctuation errors, spelling errors, inconsistencies in the use of italics were corrected.

Reviewer 4 Report

Review

A qualitative study on the health system barriers to child MMR 2and HPV vaccination among Ukrainian migrants in Poland

This interesting manuscript discusses findings about vaccine acceptance in Poland amongst Ukrainian migrants from focus groups with migrants and depth interviews with providers. It is an important and timely subject.

That said, I have some suggestions to improve the paper, both major and minor.

Major areas for improvement

You title mentions both HPV and MMR. However, the MMR vaccine is barely discussed in the manuscript (HPV is mentioned 54 times, MMR is mentioned 8). You rightly note that there is a difference between mandatory vaccines and optional ones such as HPV, but it appears that the focus is on HPV in fact. This is reasonable, but I believe that you should change the title and discuss mandatory vaccines to further highlight the challenges from HPV. In other words, I think that this is a paper primarily about vaccine hesitancy relative to HPV.

Your introduction would further benefit from more clarity that the focus of the paper is on childhood vaccination, thus convincing parents to allow their children to be vaccinated, because this is different than persuading adults to accept vaccines (such as the COVID-19 vaccine). See:

Sadaf, A., Richards, J. L., Glanz, J., Salmon, D. A., & Omer, S. B. (2013). A systematic review of interventions for reducing parental vaccine refusal and vaccine hesitancy. Vaccine, 31(40), 4293-4304.

Your inclusion of focus groups with adolescent is interesting and important. However, in the findings and the discussion, you do not highlight the differences between the findings from those focus groups and the ones with parents. I suspect that there are some very important differences that deserve more exploration.

Figure 1 is very interesting, and I understand your decision to build a framework around a WHO framework. This does lead to some limitations, nevertheless. For example, looking at your findings and Figure 1, there appears to be two general types of barriers to vaccination: resource challenges and non-resource challenges (e.g., incompatible vaccine schedules, discrimination, HPV’s status as optional, lack of trust). I mention this because the solution to resource problems is fairly straightforward (to a degree) – that is, more resources. It would be useful to further highlight the differences in your manuscript.

I believe that definitions of key concepts in academic papers are very important. You use the term “migrant” and then discuss individuals who have been in the country for several years. What is the difference for you between migrants and immigrants? How might this difference be important in terms of health provision? Do some of the individuals plan on staying in Poland versus returning to the Ukraine? A clearer definition of migrant would improve the manuscript.

I noticed that your focus groups with the migrants occurred before the Russian invasion of the Ukraine and that your provider interviews occurred before and also after the invasion. I appreciate your clarity about the dates, but this difference also deserves additional discussion. How did the invasion and influx of refugees additionally strain the healthcare system and thus the providers that you spoke to?

The discussion felt repetitive after the findings, that is, much of the discussion seemed to simply restate the findings. The discussion could be shorter and could highlight more the connection between the themes. I would focus more on recommendations (such as the paragraph that begins at line 593) in the discussion and then divide the recommendations into system-level policy recommendations (more resources, communication in schools, etc.) and provider level behavior changes (such as discriminatory behavior and/or messaging).

Minor areas for improvement

Page 4 – More description of how the coding changed over time would be beneficial, as well as including an example of how one family of codes changes and/or was merged.

The quotes from the informants are very good and appropriate, but it is hard to tell if the quotes are from the same or different informants. I would suggest adding a table with pseudonyms, age, number of children and then using the pseudonyms in the body of the document.

The inclusion of grandmothers in focus groups is interesting and presumably appropriate, but some discussion of the role of grandmothers versus mothers/fathers in vaccine decisions would be helpful. In the Ukraine, do grandmothers make many healthcare decisions for their grandchildren? Are they simply the ones taking care of the grandchildren, etc.?

A copy of the interview guides would be useful.

Line 257, the word “abandoned” is interesting. Was that word used often in the focus group?

Also, line 281, the informant says she would like more about how the vaccine would provide personal benefits – this is interesting and suggests certain messaging. Versus line 285, which implies that it is the source of the messaging that is important. So you may want to highlight content of message versus sources a way to address vaccine hesitancy.

Line 430 – are informants not interested or is cost a problem? If they can’t afford it, that is a specific barrier versus a lower priority to get vaccinated, see:

Pereira, B., Fehl, A. G., Finkelstein, S. R., Jiga‐Boy, G. M., & Caserotti, M. (2022). Scarcity in COVID‐19 vaccine supplies reduces perceived vaccination priority and increases vaccine hesitancy. Psychology & Marketing, 39(5), 921-936.

Line 586 – the wording is somewhat judgmental – the parents who believe this information do not think it is based on myth (even if it is).

Line 609 – You note that payment is not required, but several times in the manuscript you mention that parents must pay for the HPV vaccine. Again, making the focus of the paper be HPV would help clarify the cost concerns.

There are also typos throughout. For example, there should be only one space after periods and semicolons. Or line 106, it is advertisements not adverts. Also, all quotation marks should be at the top (line 232, 243, 249, etc.).

Overall, this is very interesting work! Good luck on your revision.

Author Response

This interesting manuscript discusses findings about vaccine acceptance in Poland amongst Ukrainian migrants from focus groups with migrants and depth interviews with providers. It is an important and timely subject.

That said, I have some suggestions to improve the paper, both major and minor.

Major areas for improvement

  1. You title mentions both HPV and MMR. However, the MMR vaccine is barely discussed in the manuscript (HPV is mentioned 54 times, MMR is mentioned 8). You rightly note that there is a difference between mandatory vaccines and optional ones such as HPV, but it appears that the focus is on HPV in fact. This is reasonable, but I believe that you should change the title and discuss mandatory vaccines to further highlight the challenges from HPV. In other words, I think that this is a paper primarily about vaccine hesitancy relative to HPV.

The MMR vaccine is barely discussed in the manuscript mainly because the study found that – despite healthcare system barriers - it was universally recognizable by Ukrainian migrant parents and favored by them as mandatory over the „optional” HPV vaccine. In contrast, HPV vaccine was hardly recognizable by migrants, considered less important than the MMR vaccine with high cost as fundamental barrier to vaccination. These influenced our decision to focus is on HPV.

According to the Reviewer’s suggestion, the title has been changed as follows:

Health system barriers to child mandatory and optional vaccination among Ukrainian migrants in Poland, in the context of the MMR and HPV vaccines; A qualitative study

  1. Your introduction would further benefit from more clarity that the focus of the paper is on childhood vaccination, thus convincing parents to allow their children to be vaccinated, because this is different than persuading adults to accept vaccines (such as the COVID-19 vaccine). See:

Sadaf, A., Richards, J. L., Glanz, J., Salmon, D. A., & Omer, S. B. (2013). A systematic review of interventions for reducing parental vaccine refusal and vaccine hesitancy. Vaccine, 31(40), 4293-4304. https://doi.org/10.1016/j.vaccine.2013.07.013

This was addressed as follows:

Specifically, a high influx of children refugees from Ukraine calls for urgent actions to ensure high vaccination coverage among this vulnerable population. This consequently may cause numerous public health challenges due to the need to adapt the healthcare systems to their specific requirements [10,14]. The focus of the paper is on childhood vaccination, thus convincing parents to allow their children to be vaccinated, in the context of potential barriers to immunization which parents meet and how such barriers need to be addressed. Research shows this is different than persuading adults to accept vaccines (such as the COVID-19 vaccine) [15].

[15]. Sadaf, A., Richards, J. L., Glanz, J., Salmon, D. A., & Omer, S. B. (2013). A systematic review of interventions for reducing parental vaccine refusal and vaccine hesitancy. Vaccine, 31(40), 4293-4304. https://doi.org/10.1016/j.vaccine.2013.07.013

  1. Your inclusion of focus groups with adolescent is interesting and important. However, in the findings and the discussion, you do not highlight the differences between the findings from those focus groups and the ones with parents. I suspect that there are some very important differences that deserve more exploration.

Indeed, we did not highlight the differences between the findings from those focus groups and the ones with parents due to the fact there were no significant important differences that deserve more exploration.

We added the relevant sentence in the Discussion section as follows:

Regarding the HPV vaccine there were no significant important differences observed between the findings from adolescent focus groups and the ones with parents.

  1. Figure 1 is very interesting, and I understand your decision to build a framework around a WHO framework. This does lead to some limitations, nevertheless. For example, looking at your findings and Figure 1, there appears to be two general types of barriers to vaccination: resource challenges and non-resource challenges (e.g., incompatible vaccine schedules, discrimination, HPV’s status as optional, lack of trust). I mention this because the solution to resource problems is fairly straightforward (to a degree) – that is, more resources. It would be useful to further highlight the differences in your manuscript.

We addressed this comment as follows and added it to the Conclusions section:

Interestingly, Figure 1 shows two general types of barriers to vaccination: resource challenges and non-resource challenges (e.g., incompatible vaccine schedules, discrimination, HPV’s status as optional, lack of trust. Although non-resource barriers are more challenging and time consuming, the solution to resolve resource problems is - to a certain degree - fairly straightforward. Having more health system resources available and making better use of the resources are two approaches that can suffice the needs of HCWs and meet UMs’ expectations. 

  1. I believe that definitions of key concepts in academic papers are very important. You use the term “migrant” and then discuss individuals who have been in the country for several years. What is the difference for you between migrants and immigrants? How might this difference be important in terms of health provision? Do some of the individuals plan on staying in Poland versus returning to the Ukraine? A clearer definition of migrant would improve the manuscript.

We added relevant explanation and definitions as follows:

Before the Russian aggression to Ukraine, Ukrainian migrants (UMs) in Poland were typical economic migrants. Mostly, they were working aged adults attracted by employment opportunities. In the early stages as they decide whether to stay, many fall into a pattern of migrants moving across the border and working for several months in Poland and then returning to Ukraine to extend visas before again returning to Poland. Some make a conscious decision to leave their home country and move to Poland with the intention of settling there and become immigrants.

  1. I noticed that your focus groups with the migrants occurred before the Russian invasion of the Ukraine and that your provider interviews occurred before and also after the invasion. I appreciate your clarity about the dates, but this difference also deserves additional discussion. How did the invasion and influx of refugees additionally strain the healthcare system and thus the providers that you spoke to?

We would like to thank the Reviewer for this valuable comment.

This was addressed in the Limitations section as follows:

Focus groups with the migrants occurred before the Russian invasion of the Ukraine, however, provider interviews occurred before and also after the invasion. The invasion and influx of refugees additionally strain the healthcare system and this may also influence providers that we spoke to.

  1. The discussion felt repetitive after the findings, that is, much of the discussion seemed to simply restate the findings. The discussion could be shorter and could highlight more the connection between the themes. I would focus more on recommendations (such as the paragraph that begins at line 593) in the discussion and then divide the recommendations into system-level policy recommendations (more resources, communication in schools, etc.) and provider level behavior changes (such as discriminatory behavior and/or messaging).

According to the Reviewer’s suggestions the Discussion section has been shortened and highlight more the connection between the themes.

According to the Reviewer’s suggestions, in the discussion we focused more on recommendations and divided the recommendations into system-level policy recommendations and provider level behavior changes as follows:

Several recommendations can be implemented according to this study results. System-level policy recommendations include improvement of government’ leadership and governance to recognize, assess and address the needs and priorities of UMs resulted in breaches in developing and implementing effective policies and interventions to help UMs better utilize Polish vaccine delivery system. In addition, adequate awareness raising initiatives taken by Polish policy makers are needed to improve UMs access to correct and biased information, specifically regarding vaccine importance and safety. Full reimbursement of HPV will help to reduce financial barrier for vaccination. Adequate health information systems would help with provision of credible vaccine information in translated forms, and minimize language and communication difficulties, which were experienced by both UMs and HCWs.

Our research highlights the limitation of the current national migrant registry in Poland and the need of integrating migrant status in accessible health care data.  Such policy could arm GP clinics personnel to actively manage child vaccination’ procedures. HCWs overwhelmingly agreed that without a clear protocol or guidelines on how to deliver mandatory child vaccines to migrant children, vaccination may not be offered to those in need.

The fact that our respondents mainly accessed Ukrainian sources of information, combined with the powerful anti-vaccination movement [6,9,10,21], may result in UMs in Poland being exposed to negative views about vaccination. The UMs’ population comes from the country where myths and misconceptions about non-mandatory, self-paid vaccinations are often cited by concerned parents as reasons to question having their children vaccinated [6,10]. For reasons mentioned above, providing professional information about HPV vaccine, both on the national and school level, would be of great value [32].

Provider level behavior changes are also needed. Currently, in Poland there are no courses on migrant health for GP clinics’ personnel. Such courses could positively impact unfriendly HCWs’ attitudes. In a recent study, over two-thirds of those experiencing poor treatment had a disruption in care as a result, including changing providers, delaying or forging needed care, or not following the provider’s recommendations [37]. Other studies highlighted the importance of cultural competence training as a means of eliminating racial/ethnic disparities in healthcare [38]. In this context, the  emphasis should be put on government role to ensure equal treatment of all patients presenting to vaccination services, together with a an unrestricted attitude towards UMs among HCWs.

Minor areas for improvement

  1. Page 4 – More description of how the coding changed over time would be beneficial, as well as including an example of how one family of codes changes and/or was merged.

According to the Reviewer’s suggestion, we explained this issue more thoroughly:

Interviews were collected by the research team and then analyzed thematically using the six steps defined by Braun and Clarke (2006), [21]: data familiarization, coding, theme identification (searching, reviewing and defining) and reporting. Interviews were coded using preliminary work for the RIVER-EU WP3 Delphi. The aim of the Delphi was to reach consensus, per country, on the significance of health system barriers in causing low vaccine uptake in the target populations and the feasibility of addressing these. This was done using a ranking method for both questions and with three rounds of questions. We followed the Guidance on Reporting and Conducting Delphi Study (CREDES) guidelines for the important steps and procedures to be taken in our study (Junger, 2017, Veugelers, 2020).

Thirteen barriers have been identified:

Barrier 1: Access to vaccination services is limited

Barrier 2: Language difficulties

Barrier 3: Insufficient coordination of health services

Barrier 4: Health care professionals are insufficiently trained and skilled in providing tailored care and information

Barrier 5 : Vaccine hesitancy among healthcare professionals

Barrier6 : Lack of healthcare resources

Barrier 7: Lack of or insufficient delivery of information

Barrier8: Lack of awareness raising initiatives about vaccine preventable diseases

Barrier9: Lack of government intervention to address the influence of anti-vaccination movements

Barrier 10: Vaccinations and/or additional costs are too expensive

Barrier 11: Insufficient vaccination regulations

Barrier 12: Insufficient leadership or governmental coordination in guaranteeing quality and availability of vaccination programs and promotion of these programs

Barrier 13: Insufficient governmental regulations in registering migrants for health care services, including vaccinations

For the purpose of the Polish part of the project, to enhance the rigor of the analysis, coding approaches and data interpretations were discussed between Polish team members: MG, PK and MK.

Use of these 13 barriers helped to identify where to focus policy and practice recommendations.

The themes identified were then mapped to the six WHO health systems building blocks, and the results were described according to these building blocks. Each building block contains information from the different kind of interviewees (HCWs and parents/grandparents and/or teenagers), as well as combines the vaccines enquired about, specifying whether the barriers identified are vaccine specific (i.e. only applicable to MMR and/or HPV) and more generalizable.

Each of the 13 barriers described above was mapped to one or several health system building blocks. It was therefore possible for the same barrier/enabler to appear multiple times across several building blocks.

  1. The quotes from the informants are very good and appropriate, but it is hard to tell if the quotes are from the same or different informants. I would suggest adding a table with pseudonyms, age, number of children and then using the pseudonyms in the body of the document.

We would like to thank the Reviewer for this valuable comment.

According to the RIVER-EU project regulations all personal details should be kept confidential. Therefore, together with the project’s PI a decision was made that table with participants’ coded names, age, number of children will be available on request.

  1. The inclusion of grandmothers in focus groups is interesting and presumably appropriate, but some discussion of the role of grandmothers versus mothers/fathers in vaccine decisions would be helpful. In the Ukraine, do grandmothers make many healthcare decisions for their grandchildren? Are they simply the ones taking care of the grandchildren, etc.?

This has been addressed as follows:

Grandmothers were included in focus groups after discussions with the Board of Experts in the field from the RIVER-EU project, including a Ukrainian academic professor (a pediatrician) working currently at one of the Polish universities. Thorough medical literature review was also conducted which found that, in some regions, grandmothers may serve as gatekeepers for health-seeking behavior, especially with regard to their daughters and daughters-in-law [].

Gupta, M.L.; Aborigo, R.A; Adongo, P.B.; Rominski, S.; Hodgson, A.; Engmann, C.M.; Moyer, C.A. Grandmothers as gatekeepers? The role of grandmothers in influencing health-seeking for mothers and newborns in rural northern Ghana. Glob Public Health. 2015;10(9):1078-91. doi: 10.1080/17441692.2014.1002413  

Bektas G, Boelsma F, Gündüz M, Klaassen EN, Seidell JC, Wesdorp CL, Dijkstra SC. A qualitative study on the perspectives of Turkish mothers and grandmothers in the Netherlands regarding the influence of grandmothers on health related practices in the first 1000 days of a child's life. BMC Public Health. 2022 Jul 16;22(1):1364. doi: 10.1186/s12889-022-13768-8.

  1. A copy of the interview guides would be useful.

The interview guides were the result of a collective work involving all research team members from 5 countries. Therefore, the project leaders decided that the guides will be available only for request.

  1. Line 257, the word “abandoned” is interesting. Was that word used often in the focus group?

The word “abandoned” has been changed to “uniformed” as the word used often in the FGs discussions.

’…those feeling most uninformed due to a lack of understandable, clear, official sources of information seem to rely more on word-of-mouth or social media for subsequent information on the ongoing vaccination programs”.

  1. Also, line 281, the informant says she would like more about how the vaccine would provide personal benefits – this is interesting and suggests certain messaging. Versus line 285, which implies that it is the source of the messaging that is important. So you may want to highlight content of message versus sources a way to address vaccine hesitancy.

“The only way to convince me to get vaccinated is to tell me I’ll benefit personally. If my GP  says, you know, something like: “It makes it less likely that you will get sick. Less likely 282 that you will die.” (mother, age 38, 2 children)

“Ukrainian people are difficult to convince them to get this [HPV] vaccine. We do not trust lectures and videos on the Internet too much, nor leaflets and brochures. Our GP would be the  most reliable source of information.” (mother, age 32, 1 child)

We addressed the Reviewer’s comment as follows:

UMs state they would like to know more about how the vaccine would provide personal benefits which suggests certain messaging. Additionally, source of the messaging that is important; the vast majority perceive their GP as  the  most reliable source of information about vaccination. Therefore, an adequate content of message versus reliable sources could be a useful tool to address vaccine hesitancy among UMs.

  1. Line 430 – are informants not interested or is cost a problem? If they can’t afford it, that is a specific barrier versus a lower priority to get vaccinated, see:

Pereira, B., Fehl, A. G., Finkelstein, S. R., Jiga‐Boy, G. M., & Caserotti, M. (2022). Scarcity in COVID‐19 vaccine supplies reduces perceived vaccination priority and increases vaccine hesitancy. Psychology & Marketing, 39(5), 921-936.

This is both. According to our previous study results on Ukrainian migrants in Poland

Ganczak M, Bielecki K, Drozd-Dąbrowska M, Topczewska K, Biesiada D, Molas-Biesiada A, Dubiel P, Gorman D. Vaccination concerns, beliefs and practices among Ukrainian migrants in Poland: a qualitative study. BMC Public Health. 2021 Jan 7;21(1):93. 

“they are not familiar and rather hesitant regarding self-paid vaccines. However, although none of our participants has heard about HPV vaccine, after obtaining adequate information during the focus group meeting, some would also consider it for their daughters, if the cost was refunded.”

The same conclusions come from the current study results.

  1. Line 586 – the wording is somewhat judgmental – the parents who believe this information do not think it is based on myth (even if it is).

According to the Reviewer’s suggestion, this has been changed as follows:

The UMs’ population comes from the country where myths and misconceptions about non- mandatory, self-paid vaccinations are common. Many concerned parents believe this information and do not think it is based on myth which fuels vaccine hesitancy [6,12].

  1. Line 609 – You note that payment is not required, but several times in the manuscript you mention that parents must pay for the HPV vaccine. Again, making the focus of the paper be HPV would help clarify the cost concerns.

We addressed this comment as follows and explained the system in the text:

In Poland, as in some other countries eastern and central European countries, including Ukraine, there are mandatory vaccinations which are included in the National Immunisation Programme (NIP) and provided at no cost to all children, including migrants [22]. There are also serveral vaccinations, such as HPV vaccination, recommended in the NIP. However, regarding HPV 50% of its cost has to be covered by a patient [22].

  1. There are also typos throughout. For example, there should be only one space after periods and semicolons. Or line 106, it is advertisements not adverts. Also, all quotation marks should be at the top (line 232, 243, 249, etc.).

This has been checked according to the Reviewer’s suggestions.
